# MixOmics Integration of Biological Datasets Identifies Highly Correlated Variables of COVID-19 Severity

**DOI:** 10.3390/ijms26104743

**Published:** 2025-05-15

**Authors:** Noa C. Harriott, Michael S. Chimenti, Gregory Bonde, Amy L. Ryan

**Affiliations:** 1Department of Anatomy and Cell Biology, Carver College of Medicine, University of Iowa, Iowa City, IA 52240, USA; nharriott@ucla.edu (N.C.H.); gregorybonde@uiowa.edu (G.B.); 2Department of Stem Cell Biology and Regenerative Medicine, University of Southern California, Los Angeles, CA 90033, USA; 3Hastings Center for Pulmonary Research, Division of Pulmonary, Critical Care and Sleep Medicine, Department of Medicine, University of Southern California, Los Angeles, CA 90033, USA; 4Iowa Institute of Human Genetics, Carver College of Medicine, University of Iowa, Iowa City, IA 52240, USA; michael-chimenti@uiowa.edu

**Keywords:** biomarkers, DIABLO, machine learning, multi-omics, proteomics, SARS-CoV-2, transcriptomics

## Abstract

Despite several years passing since the COVID-19 pandemic was declared, challenges remain in understanding the factors that can predict the severity of COVID-19 disease and complications of SARS-CoV-2 infection. While many large-scale multi-omic datasets have been published, integration of these datasets has the potential to substantially increase the biological insight gained, allowing a more complex comprehension of the disease pathogenesis. Such insight may improve our ability to predict disease progression, detect severe cases more rapidly and develop effective therapeutics. In this study, we have applied an innovative machine learning algorithm to delineate COVID severity based on the integration of paired samples of proteomic and transcriptomic data from a small cohort of patients testing positive for SARS-CoV-2 infection with differential disease severity. Targeted plasma proteomics and an onco-immune targeted transcriptomic panel were performed on sequential samples from a cohort of 23 severe, 21 moderate and 10 mild COVID-19 patients. We applied DIABLO, a new integrative method, to identify multi-omics biomarker panels that can discriminate between multiple phenotypic groups, such as the varied severity of disease in COVID-19 patients. As COVID-19 severity is known among our sample group, we can train models using this as the outcome variable and calculate features that are important predictors of severe disease. In this study, we detect highly correlated key variables of severe COVID-19 using transcriptomic discriminant analysis and multi-omics integration methods. This approach highlights the power of data integration from a small cohort of patients, offering a better biological understanding of the molecular mechanisms driving COVID-19 severity and an opportunity to improve the prediction of disease trajectories and targeted therapeutics.

## 1. Introduction

Since the COVID-19 pandemic ensued, a plethora of symptoms have been identified that lead to the stratification of disease severity within patients infected with SARS-CoV-2. Symptoms are akin to those observed in severe acute respiratory distress syndrome (SARS), inclusive of fever, dry cough, exhaustion, loss of taste and smell and shortness of breath [1,2,3,4,5]. The efficiency of the host’s immune response and the infectivity of SARS-CoV-2 are two core factors that define disease pathogenesis and viral survival. Despite a vast array of studies investigating the pathogenesis of COVID-19, we still do not fully comprehend the biomarkers that can predict severe disease, nor the biological pathways contributing to disease progression and severity [6,7,8,9,10,11,12]. High-throughput omics technologies have been applied to rapidly understand the mechanistic pathways of viral infection for several viruses, including dengue, zika and West Nile virus [13,14,15]. Similar large-scale multi-omic studies have been published over the past 3 years investigating the viral pathogenesis of SARS-CoV-2 [16,17,18,19,20,21,22,23,24,25,26].

As stand-alone datasets, they provide valuable information on disease pathogenesis. However, the integration of these datasets has the potential to substantially increase the depth of biological insight gained. Systems biology approaches can leverage multi-omics datasets, identify molecular biomarkers of disease and capture biological network complexity. Data Integration Analysis for Biomarker Discovery (DIABLO) is an integrative method that can be applied to identify multi-omics biomarker panels that can discriminate between multiple phenotypic groups, such as the varied severity of disease in COVID-19 patients [27,28]. An in-depth understanding of the biological changes occurring in response to SARS-CoV-2 infection can be assimilated through the evaluation of cellular and molecular features, including proteins, RNA and DNA. In this study, we detect biomarkers of severe COVID-19 using transcriptomic discriminant analysis and multi-omics integration methods. Since COVID-19 severity is known among our sample group, we can train models using this as the outcome variable and calculate features that are important predictors of severe disease. Our study highlights the power of integrating datasets to understand disease pathobiology.

## 2. Results

In this study, we examined a cohort of 70 patients, which included four independent sub-groups. These comprised the following: (1) ‘COVID-positive-ICU’, our most severe response to SARS-CoV-2 infection, with patients requiring treatment in the intensive care unit (ICU); (2) ‘COVID-positive-Inpatient’, our moderate group comprising of patients infected with SARS-CoV-2, requiring hospitalization; (3) ‘COVID-positive-outpatient’ samples, representing our mildest COVID-19 infections, where patients tested positive for SARS-CoV-2 but required no hospitalization; and (4) ‘Non-COVID-ICU’, a control group of ICU inpatients not infected with SARS-CoV-2. In this study, we refer to these cohorts as severe, moderate, mild, and negative, respectively, for simplicity. An overview of the experimental design is presented in Figure 1 and demographic information of the study cohorts can be found in [29].

### 2.1. Transcriptomic Analysis of COVID-19 Severity

To evaluate the transcriptomic changes, we used a targeted next-generation sequencing (NGS) Oncomine-Immune Response Research Assay to quantify the expression of 395 genes specifically associated with inflammatory signaling and immune-oncology research. This automated workflow allows for reproducibility across samples and requires a low RNA input, allowing for the evaluation of repeat blood samples from COVID-19 patients. Unsupervised clustering of gene expression, shown in the heatmap in Appendix A, represents the gene expression from the first sample collected from all subjects in the severe, moderate and mild COVID-19 cohorts and highlights three major clusters of gene expression representing mild/moderate (purple/red), severe/moderate (blue/red) and severe COVID-19 (blue). From the targeted Oncomine-Immune panel of the 395 genes evaluated in this study, there are 181 significant differentially expressed genes (DEGs) comparing severe to mild, 68 DEGs comparing severe to moderate, and 104 DEGs comparing moderate to mild COVID-19 cohorts (Appendix A). Among the DEGs, there are 67 genes that separate severe COVID-19 from both moderate and mild cases (Appendix A). Comparison of the transcript levels across the top 25 most DEGs between the severe and all cohorts (Figure 2A) shows four clusters of gene expression as follows: (1) genes with low expression in mild and with decreased expression with severity, (2) genes highly expressed in mild and with decreased expression with severity, (3) genes expressed in mild and highly decreased with severity and (4) genes expressed in mild and greatly increased with severity. A STRING (Search Tool for the Retrieval of Interacting Genes/Proteins) evaluation of the physical and functional protein–protein interaction network of the top 25 most significant DEGs between severe and all other cohorts (Appendix A) highlights two major clusters of predicted interactions centering around mTOR-FOXO1 signaling and PTPRC signaling. The colored nodes are proteins in our dataset, and the white nodes are proteins with predicted interactions in these networks (with a minimum interaction score of 0.7) (Figure 2B). The bar charts provide examples of severity-dependent DEGs in COVID-19 (Figure 2C–E). ARG1 and S100A9 both increase in expression with severity, with similar trends in both male and female subjects (Figure 2C). HLA-B and IFNA17 are genes that are increased with severity, but only in distinct clusters of the cohort. Interestingly, HLA-B expression is differentially associated with severity in male and female subjects, with a significantly elevated expression associated with moderate cases, specifically in females and not in males (Figure 2D). CLEC4C is an example of a DEG that is decreased with severity with exceptionally low expression in severe COVID-19 cases (Figure 2D).

### 2.2. Single ‘Omics’ Modeling with Sparse PLS-Discriminant Analysis

Next, we performed a sparse partial least squares model with discriminant analysis (sPLS-DA) that separated severe, moderate and mild COVID-19 (Figure 3A and Appendix A), but not the ‘COVID-19-positive-inpatient’ (moderate) from ‘COVID-19-negative-inpatient’ (negative) (Figure 3A). We removed the COVID-19-negative inpatient samples since they are not directly relevant to the investigation of biomarkers of COVID-19 severity; this had no impact on the ability of the model to predict COVID-19 status on RNA sequencing (RNA-seq) expression data (Figure 3B). Classification receiver operating characteristic (ROC) curves for the RNA-seq sPLS-DA model, with the COVID-19 negative samples removed, illustrate the diagnostic ability of the model with values of 0.98 for severe versus all other samples, 0.97 for moderate vs. all other samples and 1.0 for mild vs. all other samples (Figure 3C). The sPLS-DA model of the RNA-seq data alone demonstrates a nearly perfect clustering and prediction of COVID-19 severity based on just two components, comprised of 40 and 50 features in the data, respectively (Figure 3).

### 2.3. Omics’ Integrative Modeling with DIABLO

As transcriptomics and proteomics are interrelated layers of the overall system that determine a cell’s response to SARS-CoV-2 infection, we performed a multivariate analysis (described in the Methods) to integrate the changes observed in the proteomics data (described previously in [29]) and the transcriptomic data described above. We applied the Data Integration Analysis for Biomarker discovery using Latent variable approaches for Omics studies method (DIABLO) of the MixOmics package, which applies the sPLS-DA model in the context of two or more related datasets on the same set of samples. Modeling the COVID-19 positive samples demonstrated a clear separation of severe, moderate and mild COVID-19 patient samples when observing latent variables (“Components”) 1 and 2 using a weighted average of both blocks (Figure 4A and Appendix A). Introspection of the features selected by the DIABLO model enables the identification of key molecular drivers from our dataset in the context of COVID-19 severity. The top 117 features selected by the DIABLO model are reported in Appendix A. We note that the first component (‘variate 1’) primarily separates the mild samples from the rest, while the second component (‘variate 2’) partially separates the moderate from the severe. The top features (genes and/or proteins) that contribute to each of the ‘blocks’ of the first and second components (of four total components) are shown in Figure 4B,C. The top features contributing to the classification performance of the final DIABLO model along the first component in the RNA dataset include genes IL2RB, ARG1, CD4, IL10RA, CA4, MYC, CD6, CSF1R, TCF7, ZAP70, ITK, S100A9, CD8B, SIT1, as well as FCGR1A and SYND1, EN-RAGE, and WFDC-2 in the proteomic dataset. This group of genes and proteins forms a highly enriched protein–protein network (STRING analysis PPI enrichment *p*-value 1^−16^) with functional enrichments in IL-15 signaling (GO: Biological Processes), T-cell receptor complex (GO: Cellular components) and KEGG pathways for primary immune deficiency and Th1 and Th2 cell differentiation (Appendix A). Table 1 presents a summary of the published datasets that validate the severity-ranked expression of several of our first component top features, supporting the robustness of our approach in identifying features from a small patient cohort. The second component is comprised of the top-weighted genes IRS1, CCR4, TFRC, CD79A, IGSF6, SELL, MIF, IL15, CD19, CXCR2, IFITM1 and AIF1 in the RNA data and GZMB, CD70, SPARC, CD5 and LYPD3 in the proteomic dataset (Figure 4D,E). This group of genes and proteins forms a highly enriched protein–protein network (STRING analysis PPI enrichment *p*-value 2.75^−14^) with functional enrichment in the negative regulation of myeloid cell apoptosis and neutrophil activation (GO: biological processes), plasma membrane and cell surface (GO: cellular components) (Appendix A).

The circos-style plots in Figure 5A show how the selected RNA and proteomic features, having the largest loadings and cross-block correlations (in Component 1), are positively and negatively correlated to each other. The “Clustered Image Map” (CIM) heatmap shown in Figure 5B highlights the correlation strength between a given pair of features represented by the differences in “cell” color. The CIM visualizes the correlation structure extracted from both the RNA and proteomic datasets. Blocks homogeneous in color depict subsets of features from each dataset that are correlated and suggestive of a potential causal relationship. Visualizing these data as a network plot provides additional context to the correlation between features (Figure 5C). The strength of a positive (red) or negative (green) correlation between core features defining the dataset clustering is shown in the lines connecting proteins (green) and RNA (blue). The core of these correlations center around connections with Syndecan-1, EN-RAGE, WFDC2, HGF and CDCP1. Unbiased, hierarchical clustering of gene and protein expression data based on the genes used to construct the two components of the sPLS-DA model shows an almost perfect separation of severe and mild COVID-19 samples (Figure 6), highlighting the strength of this model in predicting signatures associated with COVID-19 severity.

### 2.4. GO Term and Pathway Analysis of Selected Biomarkers of COVID-19 Severity

As described in *Methods*, the final DIABLO model selected 95 and 22 features from the RNA-seq and proteomics datasets, respectively, after model tuning. These ~117 features were used as inputs for gene ontology (GO) term analysis (Table 2) and Reactome pathway analysis (Table 3) to identify the ontologies and pathways that are enriched in our COVID-19 severity biomarkers. Because we are starting from a panel of genes already selected for oncology and inflammation, we expect to see enrichment for general or high-level terms and pathways. Therefore, we only considered the GO terms and pathways, an FDR-corrected *p*-value < 0.001 and fold enrichment >10 (GO terms) as potentially significant. GO term analysis showed an enrichment in terms related to the “immune system process”, as expected. However, terms with the largest fold enrichments (>10) as well as significant FDR *p*-values were related to the regulation of “lymphocyte activation”, “T-cell activation”, “leukocyte activation” and “leukocyte proliferation”. Among the enriched pathways from ReactomeDB analysis, only four had FDR-corrected *p*-values less than 0.001 (“Interleukin-4 and Interleukin-13 signaling”, “Cytokine Signaling”, “Immune System” and “Interleukin-10 signaling”).

## 3. Discussion

By applying MixOmics to combine approaches to interrogate high-dimensional datasets, we have been able to interrogate the molecular basis of COVID-19 disease severity more comprehensively. This integrative approach contrasts with most existing studies, which only focus on a single-omics approach, which likely masks valuable information. Our data highlights the power of combined analysis of independent transcriptomic and proteomic datasets taken from the same subjects across different disease severity cohorts to elucidate complex biological mechanisms leading to severe COVID-19.

Integrative modeling discovered correlations between features previously found to be significantly differentially expressed and associated with COVID-19 severity in our proteomic data alone [29]. The application of DIABLO allowed us to identify key omics variables from our transcriptomic and proteomic datasets and was able to discriminate between COVID-19 severity cohorts (Figure 6). Importantly, the features that our model used to drive the clustering of the datasets are consistent with data from other published omics data analyses. Examples include elevated protein expression with the severity of COVID-19 for proteins, including (1) Syndecan-1 (SYND1) [36,38,48], (2) S100 calcium-binding protein A12 (EN-RAGE or S100A12) [42,49,50], (3) Hepatocyte Growth Factor (HGF) [51,52,53] and (4) CUB domain-containing protein 1 (CDCP1) [54]. Examples of elevated RNA transcript expression with the severity of COVID-19 for genes included (1) IFNA17 [55,56] and (2) ARG1 [57,58]. Interestingly, IFNA17 and HLA-B are two examples of genes where expression is associated with severity, but only in a subpopulation of patients. HLA-B is known to exhibit significant genetic diversity among individuals and will influence one’s ability to recognize and respond to viral infection by COVID-19 [59,60]. IFNA17 is an interferon that is a critical part of the innate immune response in viral infection. In support of this data, IFNA17 was discovered to be differentially expressed in a study evaluating interferon-stimulated gene profiles of post-mortem lung tissues from severe cases of COVID-19 [55]. While its expression remains an active area of research in COVID-19, it is likely that its overexpression may lead to hyperinflammation in severe COVID-19 [61,62].

While our study was designed to interrogate high-dimensional datasets where the patient sample may be limited, the small number of subjects in our study can also be viewed as a limitation. Furthermore, repeat samples were only obtained from inpatients in the USC COVID-19 Biorepository, leading to an unbalanced design of the study cohorts. This study was designed to investigate targeted panels of genes and proteins rather than taking a whole transcriptome and proteome approach. While this limits the scope of the target signatures associated with COVID-19 severity, it allows for the development of a multivariate integrative classification method that can predict signatures associated with COVID-19 severity that can be applied to integrate larger transcriptomic and proteomic datasets. The GO terms discovered through DIABLO highlighted a link between Interleukin-4 and Interleukin-13 signaling and COVID-19 severity. Il-13 was recently discovered to be a core driver of COVID-19 severity; patients prescribed Dupliumab, an antibody that blocks IL-13 and Il-4, have significantly less severe disease. This observation was backed up by data in murine COVID-19 models [63]. Indeed, IL-13 signaling has been linked to the regulation of hyaluronic acid and the persistence of post-COVID-19 conditions [63,64]. Similarly, PD-1 and the PD-L1 axis have also been connected clinically to the severity of COVID-19 [65,66,67]. PD-1 (CD279) is known to be involved in the maintenance of immune tolerance, and several studies have now reported that regulation of the PD-1/PD-L1 axis is critical in the regulation of a variety of infectious diseases [68,69]. While acutely, a reduction in infection-associated inflammation and inflammation-mediated tissue damage may be noted, chronic activation can drive immune exhaustion and be associated with increased severity of infectious diseases, such as SARS-CoV-2.

These examples and the analysis presented in this study clearly demonstrate the capacity of MixOmics to discover correlations between the features of independent datasets and generate biomarker signatures specific to disease status. A wider application of this approach to published datasets should substantially enhance our ability to identify specific biomarkers predictive of COVID-19 disease severity and assist in understanding the biomolecular pathways defining phenotype pathogenesis.

## 4. Materials and Methods

### 4.1. Patient Recruitment and Sample Collection

Patient samples were collected between 1 May 2020 and 9 June 2021 from patients seen at the Keck Hospital, Verdugo Hills, and Los Angeles (LA) County Hospital and stored in the University of Southern California (USC) COVID-19 Biospecimen Repository. At this time, no subjects were vaccinated, nor were samples analyzed for the SARS-CoV-2 variant. For this study, patients were assigned anonymized, coded IDs and were grouped according to the following cohort definitions: severe, indicating COVID-19-positive subjects who were admitted to the intensive care unit (ICU); moderate, indicating COVID-19 subjects who were hospitalized but not admitted to the ICU; mild, indicating COVID-19 subjects who tested positive for SARS-CoV-2 but did not require hospitalization; and control, indicating subjects who tested negative for SARS-CoV-2 upon admission to the ICU for treatment of other severe diseases. Population demographics for these cohorts have been previously published [29]. Participants were predominantly Hispanic/Latinx (69%), reflecting the demographics of donors available from the source biorepository (57.4% Hispanic/Latinx, https://sc-ctsi.org/about/covid-19-biorepository, accessed last on 30 June 2022).

### 4.2. Proteomics

Plasma proteomics datasets have been previously published [29]. In brief, plasma samples were analyzed by Olink proximity extension assays (PEA) (Thermo Fisher Scientific, Waltham, MA, USA) for the quantification of 184 secreted markers. Olink’s Target 96 Inflammation and Target 96 Oncology II panels were chosen for the spread of proteins related to immune response and tissue remodeling. All normalization of proteomics data was performed by Olink. Data were returned to researchers as Normalized Protein eXpression (NPX) values, which represent the signal of a given protein on a log2 scale relative to the expression of the same protein in other samples. NPX values are not comparable between different proteins; details of this methodology are shared in a white paper available at https://olink.com/knowledge/documents (accessed on 29 April 2025). Of the 184 proteins in the panels, 6 were duplicates, and 7 had NPX values under the protein-specific limit of detection (LOD) in >50% of samples in all cohorts, leaving 171 unique proteins for analysis. In total, 144 samples were analyzed. Samples were determined to fail quality control if internal incubation and detection controls deviated by a +/− 0.3 Normalized Protein eXpression (NPX) value from the median value across all samples. Four samples failed both panels and were excluded, and eight samples failed the Oncology II panel and were only included in the analysis of the Inflammation panel.

### 4.3. RNA Extraction and Quantitation

Total RNA was extracted from whole blood samples using the MagMAX for Stabilized Blood Tubes RNA Isolation Kit (Thermo Fisher Scientific), respectively, according to the manufacturer’s high-throughput protocol using the KingFisher Duo Prime Purification System (Thermo Fisher Scientific). RNA was eluted in 50 μL of MagMAX Elution Buffer, and the yield was determined by quantitative real-time PCR using the TaqMan Fast Virus 1-Step Master Mix (Thermo Fisher Scientific) and TaqMan Gene Expression Assay, GUSB (Thermo Fisher Scientific), using Promyelocytic Leukemia (HL-60) Total RNA (Thermo Fisher Scientific) as the standard, according to the manufacturer’s recommendation. A concentration of 10 ng in 7 μL, or 1.43 ng/μL, was required as an adequate yield to proceed to cDNA synthesis. Total RNA was reverse transcribed using the SuperScript VILO cDNA Synthesis Kit (Thermo Fisher Scientific) according to the manufacturer’s specifications.

### 4.4. Library Preparation and Next-Generation Sequencing

RNA libraries were prepared from reverse-transcribed cDNA samples on the Ion Chef System using the Ion AmpliSeq Kit for Chef DL8 and Oncomine-Immune Response Research Assay (Thermo Fisher Scientific). RNA libraries were immediately used for sequencing. Magnetic bead purification and size-selection steps were performed using AgenCourt AMPure XP Beads (Beckman Coulter, Brea, CA, USA) and DynaMag-PCR Magnet (Thermo Fisher Scientific). Sequencing of prepared RNA libraries was performed on the Ion Chef and Ion GeneStudio S5 Systems (Thermo Fisher Scientific). RNA libraries were sequenced using the Ion 520 and Ion 530 kits and Ion 530 chips (Thermo Fisher Scientific).

### 4.5. NGS Analysis Pipeline and QC

Base calling, alignment, read filtering and variant calling were performed on the IonTorrent Suite (v5.16) (Thermo Fisher Scientific). Reads smaller than 25 bases were removed. Thumbnail quality control reports produced by the Ion Torrent Suite were assessed for percentage ion sphere particle (ISP) loading and density, total reads and percentage usable reads, and read length. Runs were excluded if the percentage ISP loading was below 70% overall, or if the ISP density was below 50% in any region of the chip, if the percentage usable reads was below 70% or if the median read length was below 120bp. RNA libraries were aligned to the Immune Response (v3.1) reference library and analyzed using the ‘immuneResponseRNA’ plugin on the Ion Torrent Suite.

### 4.6. Single-Omics Data Analysis

An unpaired student’s *t*-test with a *p*-value ≤ 0.05 and an adjusted *p*-value (FDR) ≤ 0.1 were applied. Additionally, a fold-change cutoff of ±2 was employed to obtain the differentially expressed features. Differentially expressed proteins (DEPs) were subjected to hierarchical clustering analysis, volcano plot and Principal Component Analysis (PCA) using the Olink Statistical Analysis Application (v1.0). Gene ontology and pathway enrichment analyses were retrieved from KEGG and Reactome using g:profiler [70]. The Transcriptome Analysis Console (TAC, Applied Biosystems, v4.0.1.36) was used to perform a one-way ANOVA comparison of gene expression levels from the RNA-seq dataset. The TAC was used to perform hierarchical clustering, generate a heatmap of gene expression levels and generate volcano plots. Distances for hierarchical clustering were computed using the complete linkage method. Network analysis was performed using the STRING database (STRING Consortium, version 11.5). Protein–protein connections were assigned a combined “score” by evaluating the probabilities of interaction derived from literature and database mining and were then mapped according to these scores. The minimum required interaction score was set at the highest confidence (0.9) [71].

### 4.7. Dataset Preparation for MixOmics Analysis

Both RNA and proteomic datasets required data cleaning prior to model building. For RNA data from Ion Torrent, individual Excel sheets containing log2-scaled housekeeping normalized counts for up to 8 samples were imported into R. ‘Tidyverse’ functions were used to merge these tables into one ‘feature by sample’ matrix with the redundant gene names corrected. For proteomic data, the Olink batch-normalized data in Excel format were read into R and filtered to retain only the patient’s plasma samples. Sample name errors were fixed at this stage. The resulting data tables for RNA and proteomics were written in CSV format for use in downstream modeling.

### 4.8. Sparse PLS Modeling of RNA-Seq Data Alone

Our study included 65 matched samples with both the RNA-seq and proteomic data available; these data had 398 features (transcriptomics) and 184 features (proteomics) as inputs to the model. Both datasets were normalized according to the default software protocols, as described previously. Sparse Partial Least Squares Regression Discriminant Analysis (sPLS-DA) models on the RNA-seq data alone were calculated with and without the COVID-negative control samples included. An initial model with COVID-negative samples (N = 65) was created with feature selection disabled using ten components. A subsequent model with COVID-negative samples removed (N = 55) was subjected to performance testing with K-fold cross-validation (K = 5) and 50 repeats. The results showed the lowest error for the five components (no feature selection). Feature selection tuning was then performed with K-fold cross-validation (K = 5) and 10 repeats, using the ‘Balanced Error Rate’ (BER) and ‘max.dist’ as the measure of performance. Feature tuning showed that just two components performed as well (BER < 15%) as three or more for certain values of “keepX” (see Appendix A). The final sPLS-DA model on the RNA-seq data alone was constructed with just two components (to reduce the risk of overfitting) and selected features of 40 and 50 on each component.

### 4.9. MixOmics Multi-Omics Data Integration

The proteomic and transcriptomic datasets were integrated with ‘Data Integration Analysis for Biomarker Discovery using Latent components (DIABLO)’, a multi-omics method that maximizes the correlation between pairs of pre-specified omics datasets using sparse PLS-Discriminant Analysis. Our study included 55 matched samples (as described above) as inputs to the model. Both datasets were normalized according to the default software protocols, as described previously. A 2 × 2 matrix was used as the design matrix to tune the model towards prioritizing sample classification performance versus maximizing feature correlations (the values can range between 0 and 1) as shown in Table 4:

Model building was also explored with a range of off-diagonal values, including 0.5 (balance of classification and correlation) and 0.75 (bias toward correlation); however, we did not observe a pronounced effect on the features selected or sample classification performance. An initial DIABLO model was fit with ten components, the design matrix described above, and no feature selection for tuning and evaluation. Performance testing with K-fold cross-validation (K = 5) and 50 repeats showed that the overall balanced error rate (BER) decreased with each component until leveling out around 8 components (Appendix A). Thus, we carried eight components into tuning for feature selection with the ‘tune.block.splsda’ function (Appendix A). Tuning was performed with K-fold cross-validation (K = 5) and ten repeats using ‘centroid distance’ measures. An optimal number of features was reported for each of 8 components across both blocks, with the BER approaching ~0.1 for the optimal solutions. Performance evaluating the DIABLO model again after feature selection optimization showed that a minimum in the BER (~0.1) was now reached at only four components. We used four components in constructing the final DIABLO model, retaining (20, 25, 25, 25) and (5, 7, 5, 5) features for each of the four components in the RNA and proteomic data, respectively.

### 4.10. Gene Ontology and Reactome Pathway Analysis

Gene ontology (GO term) analysis of the features selected by the DIABLO model as significantly correlated with and predictive of COVID-19 severity for the RNA-seq (N_features = 91) and proteomics (N_features = 22) datasets was conducted using the Gene Ontology Resource Portal (geneontology.org; PANTHER v17.0, GO Database 1 July 2022). Fisher’s exact test was used for overrepresentation analysis, and an FDR correction (Benjamini and Hochberg) was applied. A total of 103 out of 110 IDs were uniquely mapped (11 were multi-mapping). The background set used for GO term enrichment analyses in Table 1 and Table 2 consisted of the set of proteins and transcripts detected in our proteomic and transcriptomic datasets rather than the entire human proteome. Pathway analysis of the same set of genes was performed with the Reactome pathway browser (reactome.org; v3.7, database release 83).

### 4.11. Study Approval

This study was approved by the institutional review board (IRB) of the University of Southern California (USC): Protocol#: HS-20-00519.

## 5. Conclusions

By leveraging the integrative capabilities of MixOmics, our study highlights the significant advantages of combining transcriptomic and proteomic datasets to uncover the molecular mechanisms driving COVID-19 disease severity. This approach reveals critical correlations between features from independent datasets, identifying biomarker signatures that distinguish severity cohorts and align with previously reported findings. This integrative analysis revealed a set of regulatory features that not only distinguish mild from severe disease but also align with key immune and inflammatory pathways implicated in COVID-19 pathogenesis.

Key proteins, elevated in severe cases, including SYND1, S100A12, HGF and CDCP1, reflect processes such as endothelial dysfunction [33,36,38,48,72,73], neutrophil activation [43,50,74,75], tissue remodeling [51,53,76,77] and systemic inflammation [54,78,79]. These findings are supported by previous studies highlighting their roles in vascular injury, immune dysregulation and barrier breakdown, all of which are central to severe manifestations of COVID-19. At the transcriptomic level, the enrichment of genes such as IFNA17 and ARG1 [57] and the involvement of pathways, including IL-13 signaling and the PD-1/PD-L1 axis, further underscore the immune-modulatory and inflammatory nature of disease progression [66,67,68].

While our study did not include in vivo validation, this reflects the recognized limitations of current animal models for COVID-19, which often require humanized ACE2 expression and still fall short of recapitulating the full spectrum of human immune responses and disease severity [80]. Additionally, acquiring new clinical samples remains a challenge, particularly from unvaccinated individuals and those with mild disease, limiting opportunities for further specimen-based validation. To address these constraints, we strengthened the robustness of our findings through comparison with publicly available datasets, which showed strong concordance with our multi-omic signatures.

Although the study was based on a small cohort and utilized a targeted panel design, the integrative methodology employed, specifically the DIABLO framework, proved effective for identifying coordinated molecular features across transcriptomic and proteomic datasets. This approach revealed biologically meaningful pathways, including immune checkpoint signaling and cytokine-mediated inflammation, that are associated with disease severity. Although the two datasets captured distinct molecular features, integrating them allowed us to identify meaningful correlations between gene expression patterns and protein abundance linked to disease progression. By leveraging the complementary information from transcriptomics and proteomics, our integrative approach provides a more comprehensive and biologically coherent view of the molecular networks driving COVID-19 severity. Several of the proteins elevated in severe cases may serve as candidate biomarkers for diagnostic or prognostic applications, while the involvement of immune-modulatory pathways points to potential therapeutic targets, particularly in patients experiencing hyperinflammatory responses. Expanding this integrative strategy to larger and more diverse cohorts holds promise for refining biomarker signatures, advancing our understanding of host–pathogen interactions, and informing precision medicine approaches to the management of COVID-19.

## Figures and Tables

**Figure 1 ijms-26-04743-f001:**
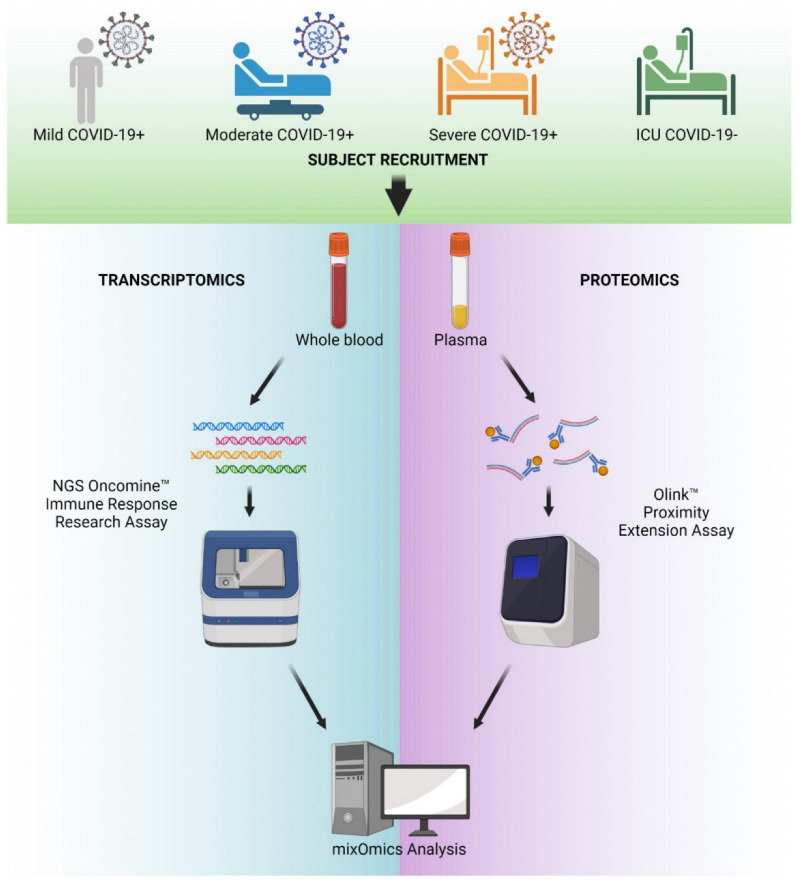
COVID-19 multi-omic study design. Schematic diagram of experimental design for the study. Blood and plasma samples were obtained from people with COVID-19 and categorized as mild, moderate, severe and ICU, but COVID-19 negative. RNA was extracted from whole blood and sequenced using the next-generation sequencing Oncomine-Immune Response target panel. Protein expression in plasma was evaluated by Olink proximity Extension Assays. The data were combined for analysis using our MixOmics platform.

**Figure 2 ijms-26-04743-f002:**
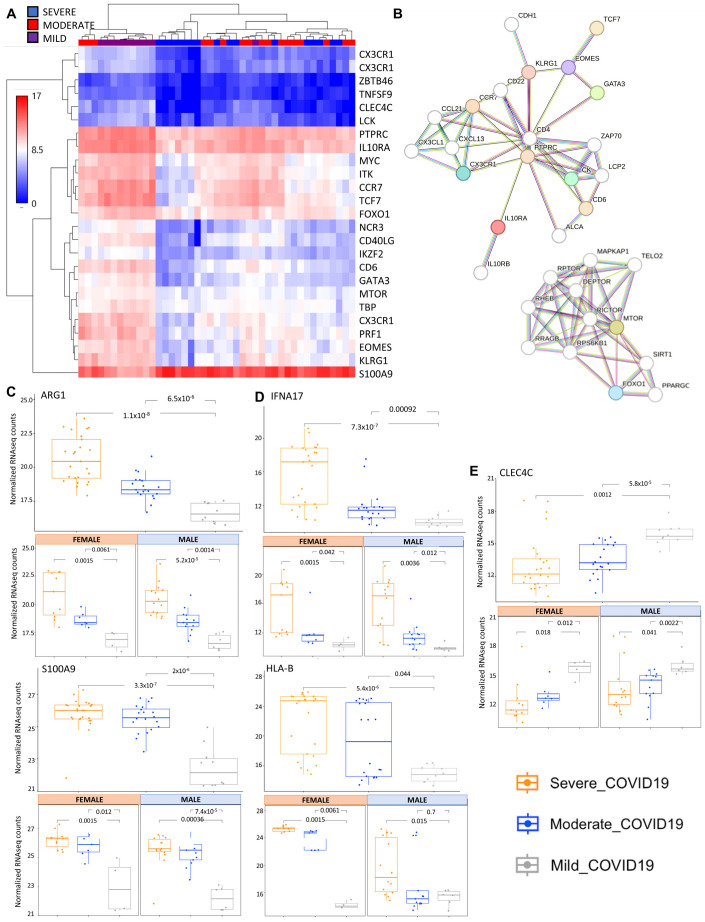
RNA-seq analysis comparing all COVID-19 cohorts. (**A**) Heatmap showing unsupervised clustering of the top 25 most significant DEGs between severe and all other COVID-19 study cohorts. Relative expression is on a scale of 0 (blue) to 17 (red) for COVID-19 cohorts severe (blue), moderate (red) and mild (purple). (**B**) STRING analysis showing predicted protein–protein interactions between the top 25 DEGs from severe COVID-19 compared to all other cohorts. Colored nodes represent query proteins, and white nodes represent the second shell of interactions. Known interactions are shown from curated databases (teal lines) or experimentally determined (pink lines). The predicted interactions shown are based on gene neighborhood (green lines), gene fusions (red lines) and gene co-occurrence (blue lines). (**C**–**E**) Bar charts comparing Log2 fold changes in average transcript level across Day 1 samples from all subjects in severe, moderate and mild COVID-19 cohorts. Examples of significant DEGs include S100A9 and ARG1, consistently elevated with severity of COVID-19 (**C**), HA-B and IFNA17, elevated with severity of COVID-19 in a portion of the subjects within the severity category (**D**) and CLEC4C, consistently decreased with severity of COVID-19 (**E**).

**Figure 3 ijms-26-04743-f003:**
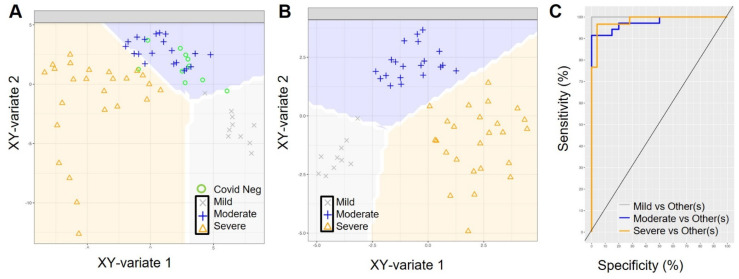
sPLS discriminant analysis for classification of COVID-19 severity. RNA-seq single dataset sPLS-DA component plots with decision background with (**A**) and without (**B**) COVID-negative samples. Samples are projected onto their XY-variate latent spaces using only the RNA-seq data and are colored by COVID-19 status. The prediction background generated by the model is plotted behind the samples, showing decision boundaries for classifying new samples. (**C**) ROC analysis of the model in (**B**) shows very high AUCs for each sample category. On the ROC plot, the diagonal line represents the randomness of the model, below the diagonal represents a poor model in its classification performance and above it represents a strong model.

**Figure 4 ijms-26-04743-f004:**
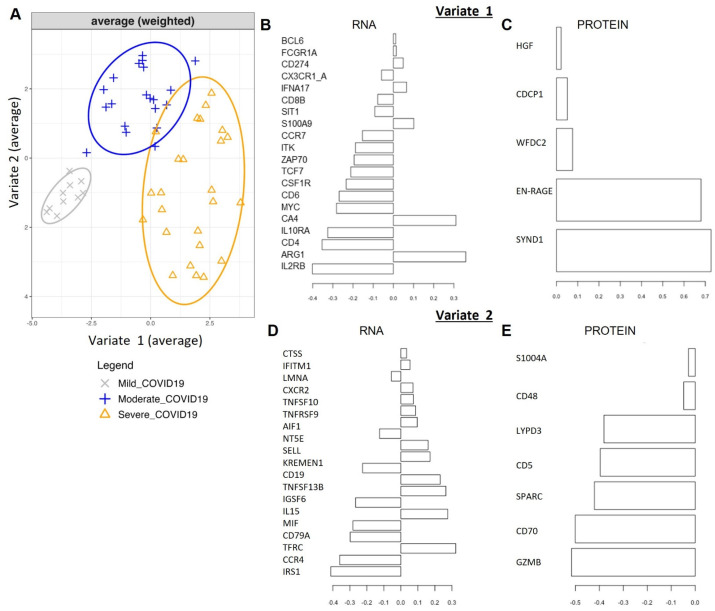
MixOmics integration of the RNA and protein datasets using DIABLO. (**A**) PCA plot of component 1 (Variate 1) and component 2 (Variate 2) of 4 components used to define sample clusters. (**B**–**E**) Top genes (**B**,**D**) and proteins (**C**,**E**) define the clustering of the samples based on component 1 (**B**,**C**) and component 2 (**D**,**E**). The *X*-axis represents the “loading” on each feature: a measure of how important it is to the trained model. This is a vector of the weight of each original variable’s contribution to the corresponding “latent” variable (Variate 1, Variate 2, etc.).

**Figure 5 ijms-26-04743-f005:**
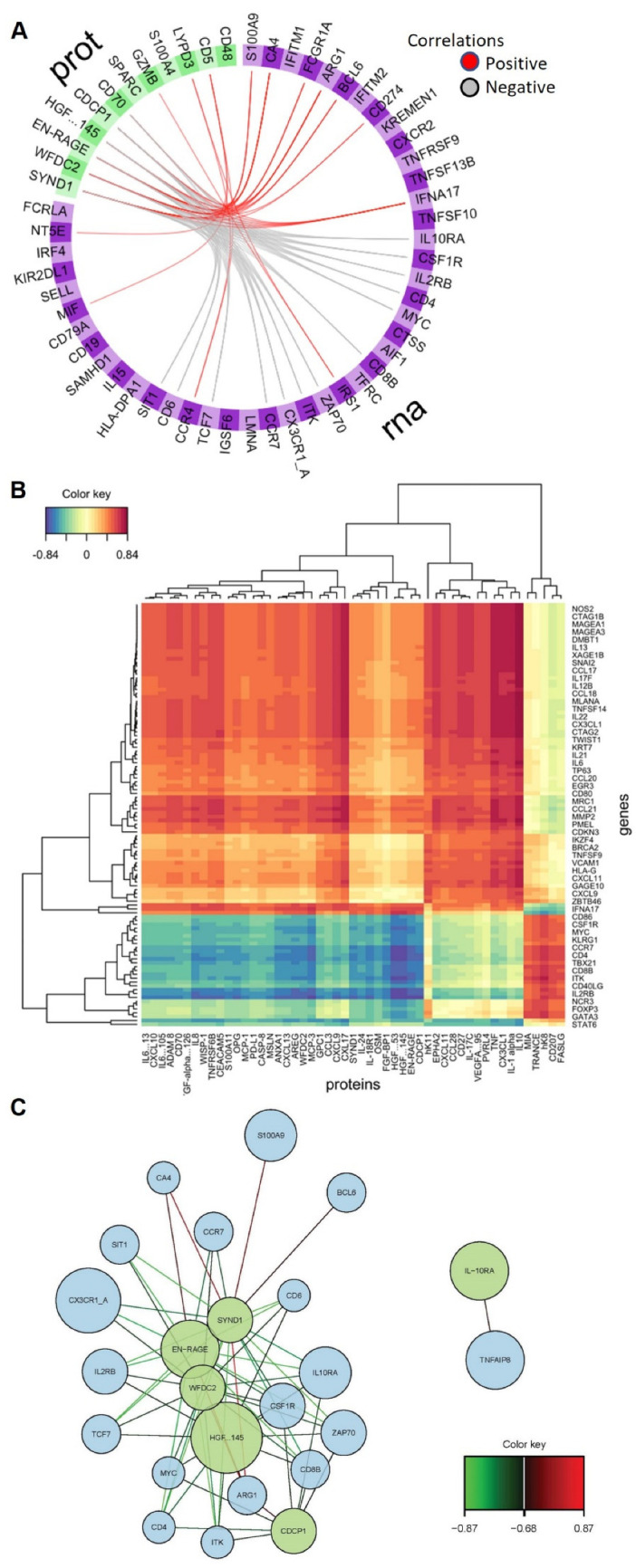
Positively and negatively correlated features of the datasets. (**A**) Circos plot showing highly positively and negatively correlated features between the RNA and protein datasets (with a correlation cutoff of 0.65). The two different datasets are segmented and colored across the circle, with each subsection representing a specific feature. The lines within the circle represent positive or negative correlations between linked variables. (**B**) Clustered expression heatmap of the highly correlated features in the DIABLO sPLS-DA model. Both features (*Y*-axis) and samples (*X*-axis) are clustered in an unsupervised manner. (**C**) Network plot of highly correlated variables where the connections represent correlations in the data (red is positive correlation and green is negative correlation). Genes are found in the blue circles, and proteins in the green circles.

**Figure 6 ijms-26-04743-f006:**
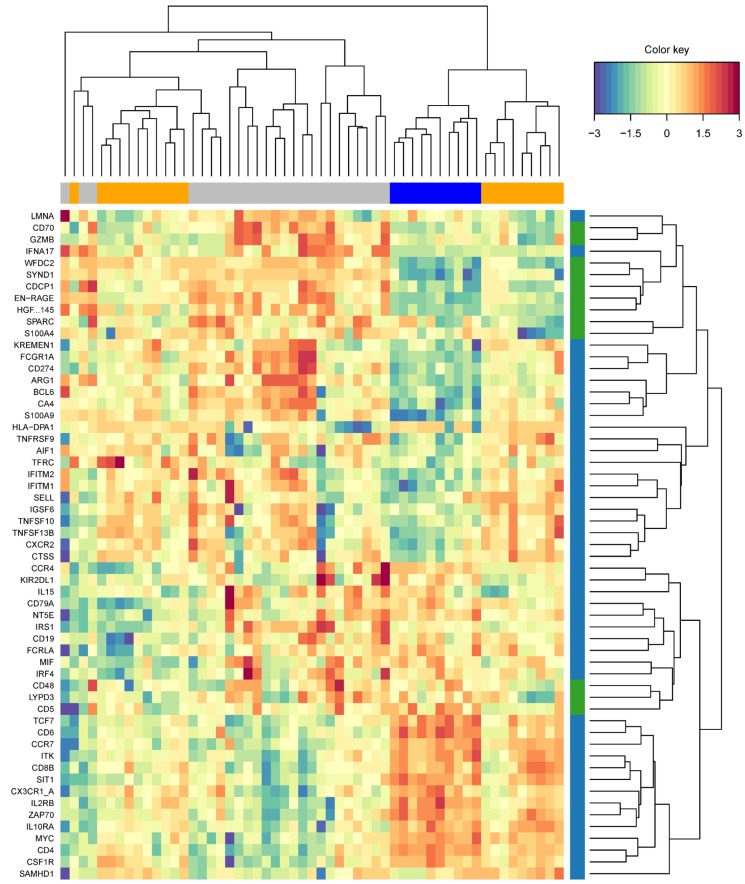
Highly correlated dataset features that maximize connections to outcomes. Heatmap generated based on analysis of all features and removing non-informative features to maximally connect highly correlated variables to the outcome. Proteins (green) and genes (blue) are depicted on the *y*-axis, and cohorts, gray (mild), blue (moderate) and orange (severe), are shown on the *x*-axis. The scale represents a relative expression from −3 (blue) to 3 (red).

**Table 1 ijms-26-04743-t001:** Top 25 GO terms enriched among DIABLO-selected features.

GO Biological Process Complete	Homo Sapiens—REFLIST (20,589)	Upload # (110)	Upload (Expected)	Upload (Over/ Under)	Upload (Fold Enriched)	Upload (Raw *p*-Value)	Upload (FDR)
immune system process (GO:0002376)	2429	73	12.98	+	5.63	1.08 × 10^−40^	1.69 × 10^−36^
regulation of immune system process (GO:0002682)	1520	56	8.12	+	6.9	1.57 × 10^−33^	1.23 × 10^−29^
immune response (GO:0006955)	1621	55	8.66	+	6.35	4.74 × 10^−31^	2.48 × 10^−27^
positive regulation of immune system process (GO:0002684)	967	43	5.17	+	8.32	5.24 × 10^−28^	2.05 × 10^−24^
regulation of lymphocyte activation (GO:0051249)	591	34	3.16	+	10.77	2.86 × 10^−25^	8.98 × 10^−22^
regulation of T-cell activation (GO:0050863)	377	29	2.01	+	14.4	8.19 × 10^−25^	2.14 × 10^−21^
regulation of leukocyte activation (GO:0002694)	684	35	3.65	+	9.58	2.00 × 10^−24^	4.47 × 10^−21^
regulation of cell activation (GO:0050865)	741	35	3.96	+	8.84	2.49 × 10^−23^	4.89 × 10^−20^
regulation of immune response (GO:0050776)	935	37	5	+	7.41	3.54 × 10^−22^	6.16 × 10^−19^
leukocyte activation (GO:0045321)	581	31	3.1	+	9.99	4.52 × 10^−22^	7.09 × 10^−19^
response to stimulus (GO:0050896)	8209	93	43.86	+	2.12	7.35 × 10^−22^	1.05 × 10^−18^
response to organic substance (GO:0010033)	2704	55	14.45	+	3.81	2.32 × 10^−20^	3.03 × 10^−17^
cell activation (GO:0001775)	700	31	3.74	+	8.29	8.05 × 10^−20^	8.41 × 10^−17^
regulation of cell population proliferation (GO:0042127)	1674	44	8.94	+	4.92	7.79 × 10^−20^	8.73 × 10^−17^
cellular response to stimulus (GO:0051716)	6569	82	35.1	+	2.34	7.34 × 10^−20^	8.86 × 10^−17^
cell surface receptor signaling pathway (GO:0007166)	2174	49	11.61	+	4.22	1.38 × 10^−19^	1.35 × 10^−16^
regulation of leukocyte proliferation (GO:0070663)	271	22	1.45	+	15.19	2.36 × 10^−19^	2.18 × 10^−16^
defense response (GO:0006952)	1478	41	7.9	+	5.19	3.50 × 10^−19^	3.05 × 10^−16^
regulation of leukocyte cell-cell adhesion (GO:1903037)	369	24	1.97	+	12.17	5.69 × 10^−19^	4.46 × 10^−16^
lymphocyte activation (GO:0046649)	465	26	2.48	+	10.47	5.54 × 10^−19^	4.57 × 10^−16^
signal transduction (GO:0007165)	4887	70	26.11	+	2.68	9.53 × 10^−19^	7.12 × 10^−16^
positive regulation of T-cell activation (GO:0050870)	253	21	1.35	+	15.54	1.10 × 10^−18^	7.84 × 10^−16^
regulation of response to stimulus (GO:0048583)	4034	63	21.55	+	2.92	4.41 × 10^−18^	3.00 × 10^−15^
positive regulation of leukocyte cell-cell adhesion (GO:1903039)	276	21	1.47	+	14.24	5.78 × 10^−18^	3.78 × 10^−15^
positive regulation of leukocyte proliferation (GO:007066)	168	18	0.9	+	20.05	6.15 × 10^−18^	3.86 × 10^−15^

**Table 2 ijms-26-04743-t002:** Reactome pathway enrichment for DIABLO-selected features.

Pathway Identifier	Pathway Name	#Entities Found	#Entities Total	#Interactors Found	#Interactors Total	Entities Ratio	Entities *p*-Value	Entities FDR	#Reaction Found	#Reaction Total
R-HSA-6785807	Interleukin-4 and Interleukin-13 signaling	20	211	3	162	0.013845	1.37 × 10^−10^	1.76 × 10^−7^	9	47
R-HSA-1280215	Cytokine Signaling in immune system	65	1115	50	2999	0.073162	4.59 × 10^−8^	2.95 × 10^−5^	290	740
R-HSA-168256	Immune System	88	2703	62	4209	0.177362	3.38 × 10^−7^	1.44 × 10^−4^	530	1659
R-HSA-6783783	Interleukin-10 signaling	11	86	2	93	0.005643	6.72 × 10^−7^	2.16 × 10^−4^	13	15
R-HSA-380108	Chemokine receptors bind chemokines	8	57	2	70	0.003740	4.74 × 10^−6^	0.001212	12	19
R-HSA-449147	Signaling by Interleukins	43	658	35	2161	0.043176	5.08 × 10^−5^	0.010863	187	505
R-HSA-389948	PD-1 signaling	5	45	1	4	0.002953	6.68 × 10^−5^	0.012223	4	5
R-HSA-202430	Translocation of ZAP-70 to Immunological synapse	5	42	3	14	0.002756	1.18 × 10^−4^	0.018894	4	4
R-HSA-9012546	Interleukin-18 signaling	3	11	1	5	7.22 × 10^−4^	2.93 × 10^−4^	0.041670	4	4

**Table 3 ijms-26-04743-t003:** Published datasets validating the expression of identified COVID-19 severity markers Syndecan-1, S100A12, WDCF2 and S100A9.

Protein	COVID Status	Sample	Data Summary	Reference
Syndecan-1	Healthy vs. COVID-19	Serum/Plasma Proteomics	Elevated	[30,31,32]
Syndecan-1	Healthy vs. COVID-19 (moderate and severe)	Plasma Proteomics	Elevated and increasing with severity	[33,34,35,36,37,38]
WFDC2	Healthy vs. COVID-19 (moderate and severe)	Plasma Proteomics	Elevated and increasing with severity	[39]
S100A12	Healthy vs. COVID-19 (moderate and severe)	Peripheral blood mRNA-seq/BAL fluid scRNA-seq	Elevated and increasing with severity	[40,41,42]
S100A12	Healthy vs. COVID-19 (moderate and severe)	Serum/Plasma Proteomics	Elevated and increasing with severity	[43]
S100A12	Healthy vs. COVID-19 (moderate and severe)	GWAS and scRNA-seq	Elevated and increasing with severity	[44]
S100A9	Healthy vs. COVID-19	PBMC protein	Elevated and increasing with severity	[45,46,47]

**Table 4 ijms-26-04743-t004:** A 2 × 2 matrix was used as the design matrix to tune the model towards prioritizing sample classification performance versus maximizing feature correlations.

	RNA	Proteomics
RNA	0	0.1
Proteomics	0.1	0

## Data Availability

The proteomics dataset has been previously published [29], and the normalized protein expression (NPX) data provided for all samples by Olink are available at https://figshare.com/s/d136a74ef05c3dfa3a21 (accessed on 29 April 2025). The Oncomine Immune Response Transcriptomic processed datasets and code to reproduce the DIABLO mixOmics analysis from data munging to model building can be found at the following link https://github.com/mchimenti/covid_multiomics_mar2022 (accessed on 29 April 2025).

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
