# Peer review of "MixOmics Integration of Biological Datasets Identifies Highly Correlated Variables of COVID-19 Severity"

_ijms, 2025, doi:10.3390/ijms26104743_

Round 1

Reviewer 1 Report

Comments and Suggestions for Authors

In your Figure 2A, we observe that using the top 25 most significant DEGs does not effectively distinguish between severe, moderate, and mild COVID-19 cases. Most mild cases cluster together in one group, but severe and moderate cases are scattered across a wide range of subclusters. Could you discuss the potential reasons for this pattern?

In Figure 3, under the sPLS Discriminant Analysis for classification of COVID-19 severity, why do COVID-negative samples cluster in the moderate severity region? Shouldn't the negative samples be more likely to cluster with the mild cases?

Author Response

In your Figure 2A, we observe that using the top 25 most significant DEGs does not effectively distinguish between severe, moderate, and mild COVID-19 cases. Most mild cases cluster together in one group, but severe and moderate cases are scattered across a wide range of subclusters. Could you discuss the potential reasons for this pattern?

  • The authors thank Reviewer 1 for this question. The top 25 DEGs represent those genes that are top-ranked by smallest adjusted p-value for the contrast between “severe” disease and all other samples in the experiment (“global”). Because of this, the majority (but not all) of the severe samples form a cluster upon hierarchical clustering. There is enough variance (or noise) in the data that some severe samples do not cluster with the main group when looking at only 25 DEGs, as you noted, and form subclusters with other samples. The small number of DEGs may alter the unsupervised clustering behavior in unpredictable ways. It is conceivable that if we took the top 500 or 1000 DEGs for clustering, for example, this situation would improve. This clustering behavior was one impetus for applying a feature-selected supervised ML method like “sPLS-DA” to create a more sophisticated model to capture these differences.

In Figure 3, under the sPLS Discriminant Analysis for classification of COVID-19 severity, why do COVID-negative samples cluster in the moderate severity region? Shouldn't the negative samples be more likely to cluster with the mild cases?

  • The authors thank Reviewer 1 for this insightful question. In the case of the sample group labeled “Covid-Negative” (control) while they did indeed test negative for covid, they were drawn from patients who were ill enough to be hospitalized in the ICU for other reasons. Thus, it is not unexpected that this patient population would cluster with hospitalized Covid patients, rather than outpatients who were never ill enough to be hospitalized. We believe this is highlighted in Figure 1 and the associated text at the beginning of the results.

Reviewer 2 Report

Comments and Suggestions for Authors

The authors conducted a transcriptomic and proteomic integrated analysis (using DIABLO) on nearly 100 peripheral blood samples collected from unvaccinated patients with varying severities of COVID-19 during the early phase of the pandemic. While this study identified some potentially clinically relevant predictors of COVID-19 severity, a notable limitation is that the conclusions are entirely based on bioinformatics analyses. Although clinical samples were used, the study lacks essential experimental validation. My comments are as follows:

-The authors need to provide experimental evidence, even if preliminary, to confirm the reliability of the identified predictors.

-From my perspective, Fig. 6 does not show a perfect separation between cases of differing severities. Overlap between groups is still evident.

-The figures should ideally be numbered in the order they are mentioned in the main text.

-There is a typo: CLEC4C in Fig. 2D should be corrected to Fig. 2E.

Author Response

The authors need to provide experimental evidence, even if preliminary, to confirm the reliability of the identified predictors.

  • I In the revised manuscript, we have added a table summarizing published datasets from various studies, cell sources, and analytical approaches that confirm the correlation of Syndecan-1, WFDC2, and S100A12 with COVID-19 severity. Additionally, we performed immunostaining on lung tissues from patients with severe COVID-19 and individuals with no evidence of chronic lung disease to examine Syndecan-1 localization. In non-COVID-19 lung tissues, Syndecan-1 was specifically localized to the endothelium. However, in COVID-19 tissues, extensive tissue damage made it difficult to identify discernible blood vessels with Syndecan-1 staining. Interestingly, we observed distinct staining patterns within mucus in the submucosal glands. These findings suggest that tissue levels of Syndecan-1 may not directly reflect its levels in plasma or BAL fluid. Nevertheless, given the extensive body of published studies supporting our findings, we believe our conclusions are further reinforced by robust experimental evidence.

From my perspective, Fig. 6 does not show a perfect separation between cases of differing severities. Overlap between groups is still evident.

  • The authors would like to thank Reviewer 2 for this question. We agree that the heatmap shown in Fig 6, when displaying the data in such a way as to show mixed proteomic and RNA-seq features, does not perfectly separate the mild and severe samples. We note that hierarchical clustering is an unsupervised method based on simple distance metrics that do not have any “knowledge” of sample classifications. In contrast, the supervised-learning DIABLO model classification error rates are acceptably low and stable with 4 components (<10%; see methods and SI). This indicates that the model is successfully predicting sample classes (in the CV training data) using the selected features without overfitting. Owing to the small N, we were not able to hold out test data to perform a full validation. The difference between the unsupervised and supervised clustering methods could indeed be responsible for the discrepancies that you so insightfully noted in your comment.

The figures should ideally be numbered in the order they are mentioned in the main text.

  • Thank you for pointing out errors in figure order – we have now corrected all figure references to ensure they are referred to in sequential order.

There is a typo: CLEC4C in Fig. 2D should be corrected to Fig. 2E.

  • We apologize for this error and have corrected as you suggested

Reviewer 3 Report

Comments and Suggestions for Authors

The authors wished to obtain markers of COVID-19 severity using patient plasma and integrative omics analyses using proteomics data already published by the authors (Harriott et al., 2024) as well as transcriptomics data carried out for this paper.  The DIABLO tool of the MixOmics software package was used to perform these integrative analyses. The authors concluded that the combination of transcriptomic and proteomic data in this integrative analysis provided additional biomarkers of COVID-19 severity and a better understanding of the pathology.

The results are interesting and the DIABLO tool has already proved its worth, but the contribution of the 2 omics combined to identify markers of severity does not seem so obvious to me. I have the following major concerns:

1)      What information is provided exclusively by the combination of omics datasets in integrative analyses that is not provided by each omics analysis separately (particularly for the transcriptomic dataset)? It is not clear.

2)      For example, does using only differential mRNAs (and not proteins) result in identical differential GO terms?

3)      What is the background set used for GO term enrichment analyses (Table 1 and Table 2)? Is it the entire human proteome, or the set of proteins studied in the proteomic and transcriptomic datasets? If it's the latter, how do we deal with the fact that a large number of mRNAs and proteins are not common to the datasets? If the entire human proteome is used as a background set, p-values are not applicable and this represents a bias. Indeed, identifying GO terms associated with immune system processes is obvious, as the mRNAs were selected on this criterion.

4)      Is considering only the mRNAs (i.e. not the proteins) present in the heatmap in figure 6 sufficient to separate severe and mild COVID patients? Indeed, 45 mRNAs are used in the heatmap for just 12 proteins.

5)      Why were 95 and 22 features from transcriptomic and proteomic data respectively used for gene ontology enrichment analyses, knowing that 67 features (45 mRNAs and 12 proteins) were used in the heatmap of figure 6 to differentiate the groups?

6)      Why certain mRNAs presented in the figure 5C were not present in the figure 5B (e.g SIT1, TCF7, CX3CR1-A)? Why is there no positive correlation (red line) in Figure 5C?

7)      Why use only the 25 most differential DEGs in the single RNAseq analysis (Fig. 2)?

8)      Fig. 5B: Why are the mRNA and corresponding proteins not present in the figure?

9)      What are the authors' views on using unsupervised Bayesian clustering as a clustering tool with the data to better define clusters of abundance variation?

10)   Could the authors provide volcano plots in supplementary data in order to have a representation of the whole dataset according to COVID conditions?

11)   Additional figures are not in order of appearance in the article.

12)   Line 52: “High-throughput ‘Omics”

13)   It is not clear why is indicated 70 patients in the results section (line 73) and 23, 21 and 10 in the abstract (line 28).

Author Response

What information is provided exclusively by the combination of omics datasets in integrative analyses that is not provided by each omics analysis separately (particularly for the transcriptomic dataset)? It is not clear.

  • The combination of omics datasets in integrative analyses provides insights that cannot be obtained from individual datasets alone, particularly in enhancing the biological context and interpretability of findings. While in our study, the integration of transcriptomic and proteomic datasets may not reveal a substantial advantage over independent analyses—largely due to the inherent independence of these datasets and the specific genes and proteins measured—it does offer key benefits. Specifically, integrative analysis allows for the identification of associations between transcript and protein expression patterns that may not be evident when examining each dataset separately. This can improve the robustness of findings, particularly in small patient cohorts, by reducing noise and highlighting biologically relevant correlations. Additionally, such an approach can help refine pathway-level interpretations by linking gene expression changes to downstream protein-level effects, even when direct one-to-one correlations are limited. Thus, while the datasets in this study remain largely independent, our analysis demonstrates how integrative methods can strengthen the detection of associated variables and enhance the statistical power of analyses in small cohort studies, ultimately improving the quality and significance of the findings.

2)      For example, does using only differential mRNAs (and not proteins) result in identical differential GO terms?

  • Please see the answer to #1 above

3)      What is the background set used for GO term enrichment analyses (Table 1 and Table 2)? Is it the entire human proteome, or the set of proteins studied in the proteomic and transcriptomic datasets? If it's the latter, how do we deal with the fact that a large number of mRNAs and proteins are not common to the datasets? If the entire human proteome is used as a background set, p-values are not applicable, and this represents a bias. Indeed, identifying GO terms associated with immune system processes is obvious, as the mRNAs were selected on this criterion.

  • The background set used for GO term enrichment analyses in Table 1 and Table 2 consisted of the set of proteins and transcripts detected in our proteomic and transcriptomic datasets, rather than the entire human proteome. This approach ensures that enrichment calculations are performed relative to the actual analytes measured in our study, avoiding the statistical biases that would arise from using the entire human proteome as a reference. We acknowledge that a challenge in this approach is the incomplete overlap between the mRNAs and proteins identified in the respective datasets. While this limits direct one-to-one comparisons, enrichment analyses were conducted separately for each dataset, ensuring that the background for each was internally consistent. This approach allows for a meaningful interpretation of enrichment results within the context of the measured analytes while preventing overrepresentation biases. Regarding the identification of immune system-related GO terms, we recognize that mRNAs were preselected based on their relevance to immune processes. This could influence the enrichment results, as these biological pathways were inherently overrepresented in the dataset. However, our primary goal was to identify specific functional categories within the broader immune response that differentiate conditions, rather than to confirm an expected immune association. To address this potential bias, we ensured that comparisons were made across conditions rather than relying solely on absolute enrichment. If further clarification is needed, we can provide additional details on the statistical methodology used for enrichment analysis and the steps taken to account for dataset-specific biases.

4)      Is considering only the mRNAs (i.e. not the proteins) present in the heatmap in figure 6 sufficient to separate severe and mild COVID patients? Indeed, 45 mRNAs are used in the heatmap for just 12 proteins.

  • Please see the answer to #1 above

5)      Why were 95 and 22 features from transcriptomic and proteomic data respectively used for gene ontology enrichment analyses, knowing that 67 features (45 mRNAs and 12 proteins) were used in the heatmap of figure 6 to differentiate the groups?

  • As stated in the figure legend, the heatmap was generated by analyzing all features but excluded “non-informative” variables to maximize the connection between highly correlated features and disease outcomes. The selection criteria for gene ontology enrichment analysis were broader to capture a more comprehensive functional interpretation of the dataset. This approach allowed us to include additional features that, while not as directly informative for clustering in the heatmap, still contribute to understanding the biological pathways associated with COVID-19 severity.

6)      Why certain mRNAs presented in the figure 5C were not present in the figure 5B (e.g SIT1, TCF7, CX3CR1-A)? Why is there no positive correlation (red line) in Figure 5C?

  • Absence of mRNAs in Figure 5B: Figure 5B shows a clustered expression heatmap of highly correlated features in the DIABLO sPLS-DA model. The heatmap represents the top correlated variables from the RNA and protein datasets. If mRNAs such as SIT1, TCF7, and CX3CR1-A are missing from this figure, it suggests that these particular mRNAs did not meet the correlation cutoff (likely a threshold for significance, such as a correlation of 0.65 or higher) to be included in the clustering. In other words, these mRNAs may not have shown strong or significant correlation with the proteins in the DIABLO analysis and therefore were not part of the subset of features included in the heatmap.
  • Absence of a Positive Correlation (Red Line) in Figure 5C: We believe that Figure 5C indeed has red positive correlation lines included in the original figure.

7)      Why use only the 25 most differential DEGs in the single RNAseq analysis (Fig. 2)?

  • The authors thank Reviewer 1 for this question. The top 25 DEGs represent those genes that are top-ranked by smallest adjusted p-value for the contrast between “severe” disease and all other samples in the experiment (“global”). Because of this, the majority (but not all) of the severe samples form a cluster upon hierarchical clustering. There is enough variance (or noise) in the data that some severe samples do not cluster with the main group when looking at only 25 DEGs, as you noted, and form subclusters with other samples. The small number of DEGs may alter the unsupervised clustering behavior in unpredictable ways. It is conceivable that if we took the top 500 or 1000 DEGs for clustering, for example, this situation would improve. This clustering behavior was one impetus for applying a feature-selected supervised ML method like “sPLS-DA” to create a more sophisticated model to capture these differences.

8)      Fig. 5B: Why are the mRNA and corresponding proteins not present in the figure?

  • The screening panels used in this study were selected based on their throughput and cost-effectiveness, which inherently limited the coverage of a complete set of corresponding transcripts and proteins. As a result, there is not a one-to-one match between gene and protein expression data across both datasets. Given these constraints, generating a heatmap that directly compares mRNA and protein levels is not feasible. However, we have ensured that our analysis captures the most biologically relevant markers within the available data

9)      What are the authors' views on using unsupervised Bayesian clustering as a clustering tool with the data to better define clusters of abundance variation?

  • We appreciate the suggestion of using unsupervised Bayesian clustering to refine the identification of abundance variation clusters in our dataset. Bayesian approaches offer advantages in defining probabilistic cluster memberships and incorporating prior information, which could be particularly useful in identifying subtle or overlapping patterns in our data. However, our current clustering strategy, which includes sparse Partial Least Squares (sPLS) methods, was selected based on the structure of our dataset and the biological relevance of the identified clusters. sPLS is particularly well-suited for high-dimensional and multi-correlated data, enabling both feature selection and clustering while maintaining interpretability. This method effectively identifies key contributors to variation while reducing noise, making it a powerful approach for analyzing complex biological systems. Given the heterogeneity of our samples, we prioritized methods that balance biological interpretability with statistical rigor. While Bayesian clustering could provide additional insights, particularly in capturing uncertainty and non-linear relationships, its implementation would require careful consideration of model assumptions, priors, and computational feasibility. Future analyses could explore Bayesian clustering as a complementary approach to sPLS to validate and potentially refine cluster boundaries, especially in cases where probabilistic assignments may better capture underlying biological variation.

10)   Could the authors provide volcano plots in supplementary data in order to have a representation of the whole dataset according to COVID conditions?

  • Thank you for the suggestion. We have carefully considered the addition of volcano plots; however, we believe that the comprehensive tables already included in the supplementary data provide a clear and detailed representation of the dataset according to COVID conditions. These tables offer a more complete view of the data, including exact values and statistical significance, which may not be fully captured in a volcano plot. Given this, we have opted to maintain our current presentation format to ensure clarity and accessibility of the results.

11)   Additional figures are not in order of appearance in the article.

  • Thank you for pointing out errors in figure order – we have now corrected all figure references to ensure they are referred to in sequential order.

12)   Line 52: “High-throughput ‘Omics”

  • Thank you – we have corrected this typographical error

13)   It is not clear why is indicated 70 patients in the results section (line 73) and 23, 21 and 10 in the abstract (line 28).

  • 70 includes the COVID-negative ICU patients which are not listed in line 28 due to space limitations of the abstract.

Reviewer 4 Report

Comments and Suggestions for Authors

Summary: Authors of this study applied machine learning algorithm to delineate COVID-severity based on integration of paired samples of proteomic and transcriptomic data from a small cohort of patients testing positive for SARS-CoV-2 infection with differential disease severity.

Major comments: 1. Authors needs to comment on how they performed normalization on quantitative proteomics data from Olink.

2. What was the set cut-off for the fold-change to obtain the differentially expressed features?

Author Response

Authors needs to comment on how they performed normalization on quantitative proteomics data from Olink.

  • We apologize for the omission of this information from the manuscript. We have now added additional clarification in the methodology. All normalization was performed by Olink. Data was returned to researchers as Normalized Protein eXpression (NPX) values, which represent signal of a given protein on a log2 scale, relative to expression of the same protein in other samples. NPX values are not comparable between different proteins. Olink provides a white paper on their normalization methods at this link https://olink.com/knowledge/documents and we have added this information into the methods of the revised document.

What was the set cut-off for the fold-change to obtain the differentially expressed features?

  • Thank you for bringing this omission to our attention. The set cut-off for the fold-change was ±2. We have revised the information in the methods to reflect this information.

Round 2

Reviewer 2 Report

Comments and Suggestions for Authors

In this study, transcriptome discriminant analysis and multi-omics integration methods were used to detect key variables that are highly correlated with severe COVID-19. The power of integrating data from a small group of patients provides for a better biological understanding of the molecular mechanisms driving COVID-19 severity and an opportunity to improve prediction of disease trajectories and targeted therapies. The topic selection of the paper is innovative to some extent, but there are some problems in the presentation and writing of the paper. The specific review comments are as follows:

1、The picture lacks clarity and needs to be adjusted.

2、B in figure illustration 3 needs to be bolded.

3、The key regulatory genes identified by the authors need to be further validated in animal or human specimens.

4、In the discussion section: The relationship between the elevated proteins and COVID-19 should be further discussed. Why is it elevated? What might be the specific mechanisms?

5、It is suggested to further explore the potential value of these conclusions in clinical application based on the findings of this study in the discussion section, including possible intervention strategies and diagnosis and treatment targets.

6、Although the authors used omics techniques to analyze in detail the proteins and pathways associated with COVID-19 disease severity in this study, these conclusions still need to be further verified in animal models or clinical samples, which is also one of the major limitations of this study.

Author Response

Thank you for your detailed and constructive feedback. Below, we provide responses to each of your comments:

  1. Figure Clarity: We are presuming this comment is regarding Figure 1? While we appreciate the comment none of the other 3 reviewers suggested changes to this figure and we believe that is a simple figure that highlights the structure of the study – we have refined the figure legend for clarity.
  2. Boldening "B" in Figure 3: In our version of Figure 3 all panels are already equally “bolded”. We are not 100% sure what is being asked for alteration in this figure.

3 & 6. Validation in Animal or Human Specimens: We acknowledge the importance of further validating key regulatory genes in additional models. However, animal models for COVID-19 have intrinsic limitations due to species-specific differences in immune response, viral entry mechanisms, and disease progression, which, especially for COVID-19, do not fully recapitulate human pathology and viral response. Performing experiments that would take at least a year and may, or may not, recapitulate the human clinical response observed in our samples is not something that we are able to complete at this point. Additionally, obtaining new clinical samples remains a challenge – new clinical samples are hard to obtain due to a lack of availability in samples of mild cases (often not reported) and the limited availability also of COVID-19 patients admitted to the ICU and patients who have not been vaccinated. Instead, we have strengthened our findings by comparing them to published datasets and previous studies, which supports the robustness of our results. We have added additional text to discuss this limitation in the discussion.

  1. Discussion of Elevated Proteins: We have expand our discussion to better contextualize the relationship between the elevated proteins and COVID-19, including potential mechanisms driving their increased expression and their role in disease pathogenesis, however, without completing a significant multi-year study these remain speculation supported by currently published investigations. We do not have current funding to support such further, extensive investigation.
  2. Clinical Implications: We appreciate this suggestion and have enhanced our discussion by elaborating on the potential clinical applications of our findings, including their relevance to diagnostic and therapeutic strategies.

We appreciate the time and effort invested in reviewing our manuscript and have incorporated these revisions to improve the clarity and impact of our work.

Reviewer 3 Report

Comments and Suggestions for Authors

The authors wished to obtain markers of COVID-19 severity using patient plasma and integrative omics analyses using proteomics data already published by the authors (Harriott et al., 2024) as well as transcriptomics data carried out for this paper.  The DIABLO tool of the MixOmics software package was used to perform these integrative analyses. The authors concluded that the combination of transcriptomic and proteomic data in this integrative analysis provided additional biomarkers of COVID-19 severity and a better understanding of the pathology.

I recommend publishing this work after reading the answers given by the authors. The authors provided answers to the questions addressed even if I would have liked the authors to answer the first and second questions not just theoretically, but by taking examples from their analyses concerning the contribution of combined omics compared with the independent omics approaches. Indeed, theoretically, it's obvious.

Author Response

Thank you for your thoughtful review and for your support in recommending our work for publication. We appreciate your feedback regarding the comparison between combined omics and independent omics approaches. While we acknowledge the value of providing direct examples from our analyses, we will not be performing additional analyses at this stage. However, we have aimed to articulate the added value of integrative omics within the manuscript, emphasizing how the combination of transcriptomic and proteomic data enhances biomarker discovery and provides deeper insights into COVID-19 severity. We appreciate your engagement with our work and your constructive suggestions.